# Loss of p53 Sensitizes Cells to Palmitic Acid-Induced Apoptosis by Reactive Oxygen Species Accumulation

**DOI:** 10.3390/ijms20246268

**Published:** 2019-12-12

**Authors:** Guowu Yu, Hongwei Luo, Na Zhang, Yongbin Wang, Yangping Li, Huanhuan Huang, Yinghong Liu, Yufeng Hu, Hanmei Liu, Junjie Zhang, Yi Tang, Yubi Huang

**Affiliations:** 1National Crop Science Experimental Teaching Demonstration Center, College of Agronomy, Sichuan Agricultural University, Huimin Road 211#, Wenjiang District, Chengdu 611130, China; 2002ygw@163.com (G.Y.); WWYYBB007@163.com (Y.W.); yangpingli103@gmail.com (Y.L.); hh820423@163.com (H.H.); huyufeng@sohu.com (Y.H.); 2Department of Regenerative and Cancer Cell Biology, Albany Medical College, Albany, NY 12208, USA; LuoH@mail.amc.edu; 3College of Science, Sichuan Agricultural University; Huimin Road 211#, Wenjiang District, Chengdu 611130, China; 03118663@163.com; 4State Key Laboratory of Crop Gene Exploration and Utilization in Southwest China, Sichuan Agricultural University, Huimin Road 211#, Wenjiang District, Chengdu 611130, China; 5Maize Research Institute of Sichuan Agricultural University; Huimin Road 211#, Wenjiang District, Chengdu 611130, China; sclydx@163.com; 6College of Life Science, Sichuan Agricultural University; Xingkang Road 46#, Ya’an 625014, China; hanmeil@163.com (H.L.); junjiezh@163.com (J.Z.)

**Keywords:** palmitic acid, p53, reactive oxygen species, cell apoptosis

## Abstract

Palmitic acid, the most common saturated free fatty acid, can lead to lipotoxicity and apoptosis when overloaded in non-fat cells. Palmitic acid accumulation can induce pancreatic β-cell dysfunction and cardiac myocyte apoptosis. Under various cellular stresses, the activation of p53 signaling can lead to cell cycle arrest, DNA repair, senescence, or apoptosis, depending on the severity/type of stress. Nonetheless, the precise role of p53 in lipotoxicity induced by palmitic acid is not clear. Here, our results show that palmitic acid induces p53 activation in a dose- and time-dependent manner. Furthermore, loss of p53 makes cells sensitive to palmitic acid-induced apoptosis. These results were demonstrated in human colon carcinoma cells (HCT116) and primary mouse embryo fibroblasts (MEF) through analysis of DNA fragmentation, flow cytometry, colony formation, and Western blots. In the HCT116 p53^−/−^ cell line, palmitic acid induced greater reactive oxygen species formation compared to the p53^+/+^ cell line. The reactive oxygen species (ROS) scavengers N-acetyl cysteine (NAC) and reduced glutathione (GSH) partially attenuated apoptosis in the HCT116 p53^−/−^ cell line but had no obvious effect on the p53^+/+^ cell line. Furthermore, p53 induced the expression of its downstream target genes, *p21* and *Sesn2*, in response to ROS induced by palmitic acid. Loss of p21 also leads to more palmitic acid-induced cell apoptosis in the HCT116 cell line compared with HCT116 p53^+/+^ and HCT116 p53^−/−^. In a mouse model of obesity, glucose tolerance test assays showed higher glucose levels in p53^−/−^ mice that received a high fat diet compared to wild type mice that received the same diet. There were no obvious differences between p53^−/−^ and p53^+/+^ mice that received a regular diet. We conclude that p53 may provide some protection against palmitic acid- induced apoptosis in cells by targeting its downstream genes in response to this stress.

## 1. Introduction

Palmitic acid, a long chain saturated fatty acid, is widely distributed in plants, including palm fruit [1,2], maize grain [3], and other cereal seeds [4,5]. In animals and humans, overloading of palmitic acid in non-fat tissues leads to lipotoxicity and diseases [6,7]. There is increasing evidence that palmitic acid induces cellular apoptosis in Chinese hamster ovary (CHO) cells [1,8,9], pancreatic β-cells [10,11], breast cancer cell lines [12], cardiac myocytes [13], vascular smooth muscle cell [14], hepatic cells [15], and neural stem cells [16]. The main pathways involved in apoptosis signaling mediated by palmitic acid are generation of reactive oxygen species [1], de novo ceramide synthesis [17], generation of nitric oxide [18], decreases in phosphatidylinositol-3-kinase [12], toll like receptor 4/reactive oxygen species (ROS)/p53 pathway [14], and modification of mitochondrial structure or function [19]. Despite many advances, the precise mechanisms have not been established and vary amongst cell types.

p53 is a multi-functional protein that plays many roles in determining cell fate in response to cellular stress [20,21,22]. Under severe stress, p53 is activated to induce cell cycle arrest, DNA repair, senescence, or apoptosis, which have been previously shown to contribute to tumor suppression [22,23]. Under stress-free or low stress conditions, p53 increases transcription of multiple downstream target genes (i.e., *p21*, *BcL2*, *Sesn1*, and *Sesn2*), engaging anti-apoptosis pathways by counteracting apoptosis in cells [22,23,24,25,26].

In this study, we investigated the role of p53 in lipotoxicity mediated by palmitic acid. Following palmitic acid treatment, apoptosis occurred in both human colon carcinoma (HCT116) cells and primary mouse embryo fibroblast (MEF) cells. Interestingly, palmitic acid induced apoptosis to a greater extent in HCT116 p53^−/−^ cells than in HCT116 p53^+/+^ cells. Likewise, MEF p53^−/−^ cells showed more apoptosis than MEF p53^+/+^ cells. An increase in p53 expression was observed following treatment with palmitic acid in p53^+/+^ cell lines. DNA laddering was used as an apoptosis marker [27]. More DNA laddering was observed 48 h after palmitic acid treatment in HCT116 p53^−/−^ cells than in HCT116 p53^+/+^ cells. Quantitative analysis of sub-G1 cells by flow cytometry [28,29] showed significantly more apoptosis after palmitic acid treatment in HCT116 p53^−/−^ cells than in p53^+/+^ cells.

To investigate the response to palmitic acid in normal cells under physiological conditions, primary MEFs were isolated and treated with palmitic acid. Primary MEF p53^−/−^ cells were sensitive to palmitic acid, like the HCT116 cell lines, and flow cytometry data showed that a substantial number of these cells became apoptotic following palmitic acid treatment. Similar results were observed for palmitic acid-treated MEF p53^−/−^ cells in the DNA laddering assay and an Annexin-V staining assay. Previous reports have shown that palmitic acid induces the formation of ROS and that is the main mechanism by which palmitic acid leads to apoptosis [3]. Therefore, ROS levels were measured in HCT116 cells and primary MEF cells. After palmitic acid treatment, ROS levels significantly increased in p53^−/−^ cell lines compared to p53^+/+^ cell lines. Under these conditions, p53 may therefore function as a scavenger of the ROS induced by palmitic acid. Realtime PCR showed that the downstream target genes *p21* and *Sesn2* were significantly induced by palmitic acid. Loss of p21 leads to more cellular apoptosis compared with wide type in the HCT116 cell line under PA stress. To gain insight into palmitic acid’s physiological effects, we constructed an animal model of obesity by feeding a high fat diet containing PA to p53^+/+^ and p53^−/−^ mice. Among mice receiving the high fat diet, glucose tolerance test assays showed higher glucose levels in p53^−/−^ mice than wild type mice; there were no differences between p53^+/+^ and p53^−/−^ mice that received the regular diet. In summary, p53 may protect cells against lipotoxicity through downstream target genes by eliminating palmitic acid-induced ROS production.

## 2. Results

### 2.1. Palmitic Acid Activates p53 in a Dose-and Time-Dependent Manner

Palmitic acid is a saturated free fatty acid that builds up in non-adipose cells, leading to cell liopotoxicity effects such as apoptosis and disease onset [30]. For example, palmitic acid can induce pancreatic β-cell dysfunction, resulting in insulin resistance and diabetes [6,30,31], and cardiac myocyte apoptosis, leading to heart failure [13,32]. p53 is a tumor suppressor, which is often viewed as a cellular guardian that protects cells from damage [33,34]. In order to investigate p53′s role in palmitic acid-induced lipotoxicity, a complex of palmitic acid and albumin with defined ratio was used to mimic saturated free fatty acids under physiological conditions. Although the normal physiological ratio of fatty acid to albumin is about 2:1, a ratio of 8:1 palmitic acid:bovine serum albumin (BSA) complex (hereafter abbreviated as PA) was used in this experimental system [1]. HCT116 p53^+/+^ and HCT116 p53^−/−^ cell lines were separately complemented with 0, 50, 100, or 250 µM PA or BSA for 24 h. p53 was mildly activated by PA in the HCT116 p53^+/+^ cell line in a dose- and time-dependent manner (Figure 1A,C). Expression levels of the important p53 target gene, *p21* (cyclin-dependent kinase inhibitor 1), also increased significantly after PA treatment (Figure 1A,C). Levels of p21 protein increased by approximately 2-fold relative to the BSA control in HCT116 p53^+/+^ cells (Figure 1B,D). Interestingly, oleic acid complexed with BSA in an 8:1 ratio (hereafter abbreviated as OA) did not activate p53 as PA did (Figure 1E,F). These results demonstrate that p53 was specifically induced by PA. Palmitic acid activates p53 in a dose- and time-dependent manner and may play an important role in lipotoxicity induced by PA.

### 2.2. Cells are more Sensitive to PA in HCT116 p53^−/−^ Cell Line Compared to HCT116 p53^+/+^ Cell Line

To further investigate p53′s role in PA treatment, HCT116 p53^+/+^ and HCT116 p53^−/−^ cells were treated with different PA doses and incubated with PA for various time periods. First, we observed that HCT116 p53^−/−^ cells became rounder and gradually broke away from the plate faster than HCT116 p53^+/+^ cells. To further observe the long-term effects of PA treatment, we seeded HCT116 p53^+/+^ and p53^−/−^ cells into 6-well plates at (4 × 10^4^) 5% confluence. After 24 h, we separately treated cells with 250 µM PA or 250 µM OA for 48 h in the wells. Crystal violet staining was then performed to visualize all cells that remained adherent to the plate under these conditions. PA treatment clearly resulted in a decrease in cell number per plate, which could reflect proliferation arrest and apoptosis, while OA treatment was ineffective (Figure 2A). To verify whether the PA effect was associated with apoptosis, we collected and extracted genomic DNA from the cells that were treated with 250 µM PA for 0–48 h. As an apoptotic cell marker, DNA laddering was observed in the gel (Figure 2B). We also extracted small fragments of apoptotic DNA, like previous results, more cell apoptosis occurred in HCT116 p53^−/−^ cells than in HCT116 p53^+/+^ cells under the same experimental conditions. Apoptotic cell was also dependent on PA treatment time (Appendix A). To quantify apoptotic cell, Sub-G1 cells were measured using flow cytometry after propidium iodide staining. After 48 h of PA treatment, the amount of apoptosis in HCT116 p53^−/−^ cells was more than 2-fold greater than that in HCT116 p53^+/+^ cells (15.5% vs. 31.1%) (Figure 2C,D). These results show that p53 may be required for cell survival under PA treatment and that knocking out of p53 leads to more apoptosis in the HCT116 cell line.

### 2.3. Distinct PA-Induced Apoptosis Effects in p53^+/+^ and p53^−/−^ Primary Mouse Embryo Fibroblast Cell Lines

To examine the physiological significance of p53 in the PA response of non-tumor cells, we isolated primary mouse fibroblasts from 13.5 days p53^+/+^ and p53^−/−^ mouse embryos. Primary MEF p53^+/+^ and MEF p53^−/−^ were treated with 250 µM PA for different time periods (0–48 h). Genomic DNA was extracted from cells, and 5 µg of DNA was separated by electrophoresis through 1% agarose gel (Figure 3A). The DNA laddering assay showed that PA induced significant apoptosis in MEF p53^−/−^ cells but only minimal apoptosis in MEF p53^+/+^ cells. Furthermore, the molecular apoptosis markers, cleaved PARP and cleaved caspase-3, were both detected by immunoblotting following PA treatment (Figure 3B). Both were significantly increased in MEF p53^−/−^ cells relative to MEF p53^+/+^ cells and p53 was slightly activated after 24 h of PA treatment (Figure 3B and Appendix A). This result demonstrates that slightly activated p53 is enough to at least partly inhibit PA-induced apoptosis in MEF cells. Similarly, Sub-G1 cells were measured by flow cytometry after staining with propidium iodide. After 48 h of PA treatment, PA-induced cell apoptosis rose from 1.18% to 13.7% in MEF p53^+/+^ and from 2.43% to 43.5% in MEF p53^−/−^ (Figure 3C).

Phosphatidylserine is translocated from the inner to the outer leaflet of the plasma membrane in the earliest stages of apoptosis and annexin V binding to phosphatidylserine is a strong and specific signal for this early apoptotic event [35,36]. We performed an annexin binding essay on MEF cells after 24 h of PA treatment. The results showed more MEF p53^−/−^ cells entering apoptosis than MEF p53^+/+^ cells (Figure 3D). Like the results seen in HCT116 cells, these results demonstrate that even mildly activated p53 may be enough to at least delay or repair cell damage induced by PA treatment and thus avoid cell apoptosis.

### 2.4. ROS Contributes at Least Partly to the Cell Apoptosis Induced by PA

Accumulation of ROS in cells treated with PA is known to be a contributing factor in PA-induced lipotoxicity [37,38,39,40]. Therefore, we used H_2_DCFDA to test whether p53-mediated anti-apoptosis was associated with a decrease in ROS. DCFDA diffuses into cells, where it is deacetylated to DCF, which in turn fluoresces upon reaction with a variety of ROS [41]. ROS was measured in HCT116 p53^−/−^ and p53^+/+^ cells treated with 250 or 500 µM PA for 24 h. As Figure 4A,B shows, ROS accumulated more in HCT116 p53^−/−^ cells than in HCT116 p53^+/+^ cells. ROS increased almost 4-fold compared with BSA controls in HCT116 p53^−/−^ cells from the PA500 treatment group; however, ROS increased only 2-fold compared with BSA controls in HCT116 p53^+/+^ cells from the same treatment group (Figure 4A,B). Similarly, we measured ROS levels in MEF cells treated with PA500; ROS increased 2-fold in MEF p53^−/−^ cells but increased only slightly in MEF p53^+/+^ cells (Figure 4C). Treatment for 24 h with PA and the ROS scavengers N-acetyl-cysteine (NAC) or glutathione (GSH) decreased DNA laddering observed in MEF p53^−/−^ cells but not MEF p53^+/+^ cells, suggesting that NAC and GSH partially reduced PA-induced apoptosis in MEF p53^−/−^ cells (Figure 4D). These results showed that ROS play an important role in apoptosis and that HCT116 and MEF cells are at least partly dependent on p53 to eliminate ROS production in response to PA treatment.

### 2.5. Gene Expression Induced by p53 Under PA Stress in HCT116 and MEF Cell Lines.

To study potential mechanisms of PA stress, we examined the expression of two key genes induced by p53: *p21* and *Sesn2*. *p21* expression has been widely investigated in response to different stresses. HCT116 cells and MEF cells were treated with 250 μM PA for different time periods. Real-time fluorescence quantitative PCR was performed to measure the expression of *p21*. *p21* expression was induced 2- to 3-fold higher by PA in p53 wild type cells (Figure 5A,C). Sesn2 plays an important role in maintaining redox and metabolic homeostasis, previous studies have reported that Sesn2 is involved in antioxidant processes as a downstream target of p53. We measured the expression of *Sesn2* in response to PA treatment of p53^+/+^ and p53^−/−^ cell lines. After 36 h of PA treatment, *Sesn2* expression increased more than 3-fold in HCT116 p53^+/+^ cells but only 2.8-fold in HCT116 p53^−/−^ cells. Similarly, *Sesn2* expression increased more than 2.5-fold in MEF p53^+/+^ cells but only 1.6-fold in MEF p53^−/−^ cells after 36 h of PA treatment (Figure 5B,D). In order to further investigate the effect of p21 under PA stress, we detected cell apoptosis in the HCT116 p21^−/−^ cell line. The results indicate that loss of p21 is more sensitive for cells to palmitic acid-induced apoptosis compared with HCT116 p53^−/−^ and HCT116 p53^+/+^ (Figure 5E). These results show that p53 may play a protective role in PA stress through the expression of its target genes.

### 2.6. Biological Effect on Mice in PA Stress

Previous investigations have reported that palmitic acid induces pancreatic β-cell dysfunction, resulting in insulin resistance and diabetes. To further study the potential biological significance of p53 in PA stress, we chose approximately three-week-old littermates of p53^+/+^ and p53^−/−^ mice to receive high fat food or regular food for 52 days, followed by performance of glucose tolerance test assays. After 52 days, p53^+/+^ mice that received the high fat diet had increased from 18.55 to 44.03 g in weight, and p53^−/−^ mice that received the high fat diet had increased from 16.49 to 40.79 g. On the other hand, p53^+/+^ mice that received regular food increased from 16.59 to 29.21 g, and p53^−/−^ mice that received regular food increased from 15.85 to 27.75 g. These results show that the body weights of p53^+/+^ mice and p53^−/−^ mice increased to different extents. p53^−/−^ high fat diet mice were significantly heavier compared with p53^−/−^ regular food mice and p53^+/+^ high fat diet mice were significantly heavier compared with p53^+/+^ regular food mice. Clearly the high fat diet leads to obesity and the model of obesity was successfully constructed. However, p53^−/−^ high fat diet mice were not significantly different in body weight from p53^+/+^ high fat diet mice. Nor were p53^−/−^ regular food mice different in weight from p53^+/+^ regular food mice (Figure 6A). In order to further demonstrate potential biological significance of p53 in PA stress, an intraperitoneal glucose tolerance assay was performed according to previously reported protocols [42,43]. Glucose values were significantly different between p53^−/−^ high fat diet mice and p53^+/+^ high fat diet mice 15 min after injection of glucose. However, there were no differences in glucose values between p53^−/−^ and p53^+/+^ regular food mice (Figure 6B). These results indicate that glucose resistance was reduced in p53^−/−^ high fat diet mice than in p53^+/+^ high fat diet mice.

## 3. Discussion

Activation of p53 signaling under stressful conditions is widely accepted to mediate suppression of tumor growth by triggering senescence or apoptosis, depending on the type/amount of stress as well as the genetic background of the cells [44,45,46]. Interestingly, it was reported when p53′s ability to induce apoptosis, cell cycle arrest, and senescence was blocked, it was still able to suppress tumor formation [47]. Therefore, p53′s other functions, such as suppression of glucose uptake and regulation of glycolysis and ROS, are also important for inhibition of tumor formation [48,49]. More and more evidence has been proposed that p53 regulates a large number of genes involved in cell metabolism, such as *PTEN, TIGAR, SCO2, GAMT, SCO2*, and *Lpin1* [23]. In addition, there is increasing evidence that p53 can also function to prevent apoptosis [50,51,52]. The level of p53 expression is the key factor in determining cell fate [52]. In our current research, we found that low or normal levels of p53 had an anti-apoptotic and pro-survival effect in response to palmitic acid. When we knocked down the expression of *p53* by siRNA and treated these cells with PA, we could not detect increased apoptosis relative to wild type. When we over-expressed *p53* in HCT116 cell lines even without PA, we detected more cell apoptosis compared to cells expressing normal *p53* level. Therefore, base level or low level of *p53* is important for apoptosis induced by PA.

It has already been shown that several cell types undergo apoptosis in response to palmitic acid treatment, including pancreatic β-cells [10,11], breast cancer cell lines [12], cardiac myocytes [13], vascular smooth muscle cells [14], hepatic cells [15], and neural stem cells [16]. These apoptotic systems are associated with various diseases. For example, pancreatic β-cell dysfunction leads to insulin secretion resistance which in turn causes hyperglycemia and diabetes. Likewise, apoptosis of cardiac myocytes results in heart failure. Here, more apoptosis occurred in MEF p53^−/−^ cells compared with MEF p53^+/+^ cells, which is important because the absence of p53 may contribute to various apoptosis-related diseases. Our results suggest that p53 can potentially promote cell survival under PA treatment. Consistent with the known functions of p53, p53 protects cells against DNA damage under low-level cellular stress in normal cells [34].

Accumulation of ROS in cells treated with PA has been shown to be a contributing factor in PA-induced apoptosis [1]. Increasing evidence shows that p53 regulates ROS through various pathways in vivo and in vitro [53]. p53 generates ROS under high cellular stress conditions and after severe cellular damage inducing apoptosis [51]. Interestingly, under normal or low stress states, p53 can also promote cell survival through p21 induction of cell cycle arrest and recruitment of DNA repair factors such as p53R2 [54] and GADD45 [55]. In a previous report, the expression of *GADD45* in the HepG2 cell line increased following treatment with PA [56]. In our current research, p53 was mildly activated in a dose- and time-dependent manner in the HCT116 cell line following PA treatment. p53 was activated potentially to promote *GADD45* expression, echoing research in HepG2. The apoptosis response of p53^−/−^ cells was more than 2-fold that of p53^+/+^ cells. At same time, ROS production increased to a greater extent in HCT116 p53^−/−^ cells than in HCT116 p53^+/+^ cells. The antioxidants glutathione and N-Acetyl cysteine at least partly attenuated apoptosis caused by palmitic acid in HCT116 p53^−/−^ cells but had no obvious effect in HCT116 p53^+/+^ cells. This result shows that some functions of p53 may have antioxidant effects. As a downstream target of p53, the antioxidant-related gene *Sesn2* increased in expression after PA treatment to a greater extent in HCT116 p53^+/+^ cells than in HCT116 p53^−/−^ cells. It partly abolished ROS in HCT116 p53^+/+^ cells but had no obvious effect in HCT116 p53^−/−^ cells. Therefore, the potential mechanism by which p53 protects cells from apoptosis is by inducing *Sesn2* gene expression, partially ameliorating ROS damage to cells after PA treatment.

As an important p53 response gene, *p21* encodes an inhibitor of cyclin-dependent kinases [40], whose expression is sensitive to even low levels of p53 activation. The expression of *p21* leads to cell cycle arrest at G1 or G2, which temporarily allows cells to survive until damage has been repaired or cellular stress resolved [57]. We compared the expression of *p21* in HCT116 p53^+/+^ and HCT116 p53^−/−^ cells under PA treatment. Although p53 was moderately induced by PA treatment, the expression of *p21* in HCT116 p53^+/+^ cells was more than 3-fold than in HCT116 p53^−/−^ cells after 36 h of PA treatment. The level of ROS was also significantly higher in HCT116 p53^−/−^ cells compared to HCT116 p53^+/+^ cells. Furthermore, more apoptosis occurs in HCT116 p21^−/−^ cells compared with HCT116 p53^−/−^ and HCT116 p53^+/+^ cells under the same PA stress. Although we have no evidence to show that p21 can directly reduce ROS levels, our data show that more apoptosis was detected in cell lines that also exhibited relatively low p21 levels. This observation is consistent with multifunctional property of p21, which includes its ability to suppress apoptosis by acting at different levels of the death cascade, and to promote growth arrest through senescence [44,58]. Therefore, cell cycle arrest was induced by p21, which at least partly contributed to p53-mediated survival under PA stress. In the current work we used standard molecular and cellular features of apoptosis. However, it is now well known that cells triggered to undergo apoptosis can recover from brink of death through a process called anastasis [59]. Therefore, to what extent the responses reported herein reflect death remains to be determined. In addition, glutathione and N-Acetyl cysteine could not completely abolish apoptosis under PA treatment in both HCT116 and primary MEF cell lines. There may be other mechanisms by which p53 contributes to cell apoptosis protection in response to PA treatment.

Physiologically, p53 appears to act as cell protector in vivo under PA stress. In our mouse model of obesity, we observed higher glucose level in p53^−/−^ mice that received a high fat diet than in p53^+/+^ mice that received the same diet. Although we did not directly measure β-cell dysfunction in vivo, we hypothesize that β-cell apoptosis occurs to a greater extent in p53^−/−^ mice than in p53^+/+^ mice under PA stress. Overall, this research highlighted differences in the apoptotic effect induced by PA in p53^+/+^ and p53^−/−^ cell lines. Our results suggest that p53 can potentially protect cells from apoptosis induced by PA treatment.

## 4. Materials and Methods

### 4.1. Materials

Palmitic acid, oleic acid, and N-Acetyl cysteine (NAC) were purchased from Fisher Scientific. Fatty acid-free bovine serum albumin (BSA), propidium iodide (PI), and glutathione were purchased from Sigma. 5-(and-6)-chloromethyl-2′,7-dichlorodihydrofluorescein diacetate and acetyl ester (Cat: C6827) were purchased from Invitrogen. The Annexin V-FITC Fluorescence Microscopy kit (Cat: 556547) was obtained from BD Pharmingen.

### 4.2. Preparation of Albumin-Bound Fatty Acid

We dissolved 0.046 g palmitic acid in 0.01 M NaOH, for a total volume of 15 mL, by stirring at 70 °C for 30 min. Dropwise addition of 1 N NaOH facilitated solubilization of the palmitic acid. It was immediately added to 2× PBS albumin solution dropwise over 30–40 s while stirring. When the palmitic acid solution became clear, the solution was adjusted with 0.1 N HCl to pH 7.4. The resulting solution had an 8:1 molar ratio of palmitic acid to BSA. Albumin-bound palmitic acid solution (palmitic acid) was diluted 12-fold in cell culture medium. An albumin-bound oleic acid solution was also prepared as above; 6 mM oleic acid was used as stock solution. We dissolved 56.7 µL of pure oleic acid in 15 mL PBS; the clear solution was immediately added to the PBS albumin solution dropwise over 30–40 s while stirring. These methods were adapted from Spector’s Protocol [60].

### 4.3. Cell Culture and Isolation of Mouse Embryonic Fibroblasts

HCT116 cell lines were from Bert Vogelstein lab and cultured in McCoy’s 5A medium (Cellgro) supplemented with 10% FBS, 50 units/mL penicillin, and 50 units/mL streptomycin. Primary mouse embryonic fibroblasts were isolated from embryos of p53^+/−^ mice at 13.5 days post fertilization and cultured in Dulbecco’s modified Eagle’s medium supplemented with 10% FBS, 50 units/mL penicillin, and 50 units/mL streptomycin. For palmitic/oleic acid treatment, cell culture medium was supplemented with palmitic acid or oleic acid both complexed with BSA at an 8:1 molar ratio. All cells were cultured in a humidified atmosphere of 5% CO_2_ at 37 °C.

### 4.4. SDS-PAGE and Immunoblotting

Cells were washed with cold PBS and lysed in flag lysis buffer (50 mM Tris-Cl pH 7.3, 137 mM NaCl, 10 mM NaCl, 10% glycerol, 0.5 mM EDTA, 1% Triton X-100, 0.2% Sarkosyl and fresh protease inhibitor cocktail from Sigma, Cat: P8340). Protein concentration was determined by the Bio-Rad protein assay (Cat: 500-0006). For immunoblotting, proteins were resolved by SDS-PAGE and transferred to Amersham Hybond ECL Nitrocellulose membranes (GE healthcare, Cat: RPN303B), followed by incubation with primary and secondary antibodies and detection with the ECL kit (GE Healthcare, Cat: RPN2109). All Western blot assays were performed independently at least three times. The following antibodies were used in this study: p53 (DO-1), p21 (SC-19) from Santa Cruz; caspase-3, PARP from Cell Signaling; and β-actin from Sigma.

### 4.5. Cell Viability Analysis and ANNEXIN V-FITC Staining

To measure cell apoptosis rates, cells were seeded into 6-well plates at (4 × 10^4^) 5% confluence and then incubated with fatty acids for 24 h. After removing the medium, colony staining solution (0.5 g crystal violet, 27 mL 37% formaldehyde, 100 mL 10× PBS, 10 mL methanol, 863 mL H_2_O) was added to cover the plate. Plates were incubated for 20 min at room temperature, washed with water, and air dried. Pictures were obtained with an Epson Perfection V700 Photo scanner once the plates had dried. Annexin V-FITC binding was performed according to the recommended protocol. Cells were seeded in 6-well plates, and after 24 h they were treated with 500 mM palmitic acid for 24 h. The cells were briefly washed twice with 1× PBS, washed once with 1× Annexin V binding buffer and then stained with Annexin V-FITC diluted 1:10 in 1× Annexin V binding buffer for 15 min at RT. After staining, the cells were washed once with 1× binding buffer; then binding buffer was added for observation and photography with an Advanced Microscopy Group EVOS f1 microscope.

### 4.6. Propidium Iodide (PI) Staining and Flow Cytometry.

Cells were treated with 500 µM palmitate. They were then washed with PBS, trypsinized, collected at 0, 12, 24, 36, and 48 h, and suspended in PBS. Cells were fixed with cold methanol overnight, centrifuged at 3000 rpm for 5 min, spun down, suspended in PBS with 20 µg/mL RNase for 30 min at 37 °C, stained with 2 µg/mL propidium iodide for 10 min and analyzed for DNA content by flow cytometry. Cells were treated with 500 µM palmitate. They were then washed with PBS, trypsinized, collected at 0, 12, 24, 36, and 48 h and suspended in PBS. Cells were fixed with cold methanol overnight, centrifuged at 3000 rpm for 5 min, spun down, suspended in PBS with 20 µg/mL RNase for 30 min at 37 °C, stained with 2 µg/mL propidium iodide for 10 min and analyzed for DNA content by flow cytometry.

### 4.7. ROS Measurement and DNA Ladder Assay

To determine ROS levels after supplementation with fatty acids at different time points, HCT116 cells and primary MEF cells were stained for 15 min with fresh 2 µM H_2_DCFDA (Invitrogen) dissolved in DMSO. Cells were then washed with HBSS buffer and analyzed by flow cytometry. HCT116 cells and primary MEF cells were harvested, including floating cells, at various time points after supplementation with 500 µM palmitic acid or 500 µM palmitic acid with 200 µM N-Acetyl cysteine (NAC) or glutathione. Cells were suspended in 475 µL of lysis buffer (10 mM Tris, pH 8.0, 100 mM NaCl, 10 mM EDTA, 0.5% SDS) with freshly added protease K (25 µL of 20 mg/mL) and incubated overnight at 50 °C. Then, 5 M NaCl was added dropwise to give a final concentration of 200 mM and the samples were purified using the genomic DNA isolation protocol for the Manual Phase Lock Gel (PLG) kit (5Prime). Equal quantities (5 µg) of each DNA sample were run on 1.0% agarose gels in TAE buffer (40 mM Tris-Cl pH 8.0, 20 mM acetic acid, and 1 mM EDTA). Images were collected with a BioDoc-IT Imaging system.

### 4.8. Construction of a Model of Mice Obesity and Glucose Resistance Assay 

Mice (C57BL/6J) were maintained in the Animal Core Facility following procedures approved (approval number pro2012426, 10th, April, 2012) by the Animal Care and Use Committee of Albany Medical College. Five littermates of p53^−/−^ and p53^+/+^ mice were chosen to receive either a high fat diet (Research Diets, Inc., Cat: D12492) or regular food (Research Diets, Inc., Cat: 12450B) after genotyping at 21 days old. Each group contained five p53^−/−^ or p53^+/+^ mice that received a high fat diet or a regular diet for 52 days. An intraperitoneal glucose tolerance test (IPGTT, 1 g glucose/kg body weight) was conducted on fasted mice (for 12 h). Glucose levels were measured with the OneTouch Ultra glucometer (Lifescan Benelux, Beerse, Belgium) in 0, 15, 30, 45, 60, 90, and 120 min by tail vein puncture. The statistical significance of differences between means was assessed using Student’s *t*-test (** *p* < 0.01; * *p* < 0.05). Data were expressed as mean ± SE, *n* ≥ 3.

### 4.9. Real Time PCR and Statistical Analysis

RNA was extracted from cells using TRIZOL Reagent (Life Technologies, Cat: 15596018) and cDNA was prepared using MMLV reverse transcriptase according to the relevant protocol (NEB, Cat: M0253L). Quantitative real time PCR was performed with Fast SYBR Green QPCR master mix (AB Applied Biosystems, Cat: 4367659) using the following primers: *p21* forward: 5′-CCATGT GGACCTGTCACTGTCTT-3′, *p21* reverse: 5′-CGGCCTCTTGGAGAAGATCAGCCG-3′; *Sesn2* forward: 5′-TCCGCCACTCAGAGAAGTC-3′, *Sesn2* reverse: 5′-GTTCAGGAAGGCCACAACAC-3′. The statistical significance of differences between means was assessed using Student’s *t*-test (** *p* < 0.01; * *p* < 0.05). Data were expressed as mean ± SE, *n* = 6.

## 5. Conclusions

We identified differences in PA-induced apoptosis between p53^+/+^ and p53^−/−^ cell lines. p53 potentially protects cells from PA-induced apoptosis though induction of its target genes, *p21* and *Sesn2*. p21 induces cell cycle arrest, and Sesn2 functions as an antioxidant factor in response to reactive oxygen species induced by PA in p53 wild type cell lines. Expression of *p21* and *Sesn2* was higher in p53 wild type cell lines than in p53 null cell lines and loss of p21 leads to more apoptosis in HCT116 compared with HCT116 p53^−/−^ and p53^+/+^ cell line, implying that p53 serves a protective function. In vivo, we fed a high fat diet to p53^+/+^ and p53^−/−^ mice to create a model of obesity. We observed greater glucose resistance in p53^−/−^ high fat diet mice than in p53^+/+^ high fat diet mice. p53 potentially protects cells from apoptosis induced by PA stress. This study may open a door to the assessment of healthy diet and p53-related tumor suppression.

## Figures and Tables

**Figure 1 ijms-20-06268-f001:**
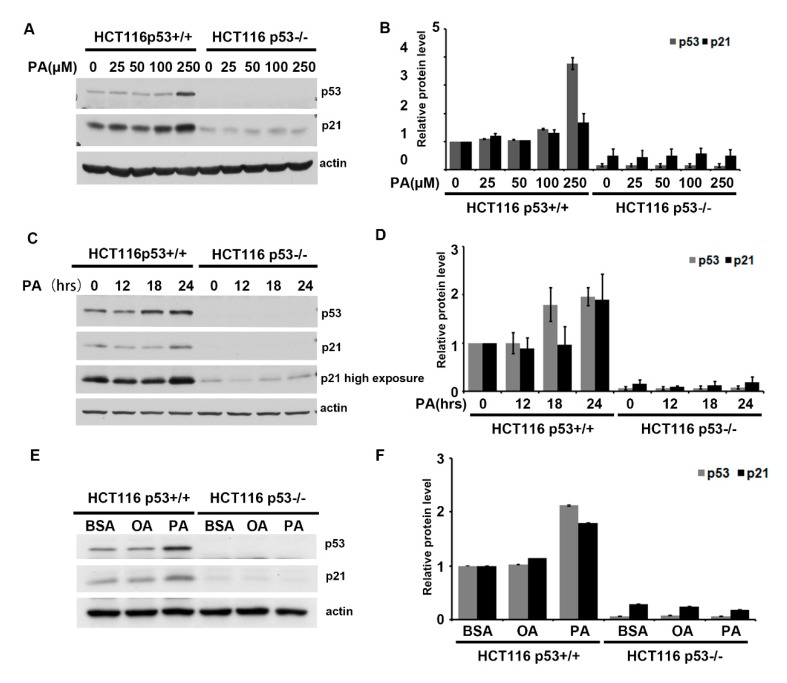
Palmitic acid induces p53 expression in human colon carcinoma cells (HCT116) in a dose- and time-dependent manner. (**A**) HCT116 p53^+/+^ and p53^−/−^ cell lines were treated with the indicated dose of palmitic acid. A total of 40 µg of total protein extract was resolved on SDS-PAGE. Immunoblotting was performed using p53 and p21 antibodies, and β-actin was used as a loading control. (**B**) Quantitative analysis of (A) by ImageJ software. (**C**) HCT116 cells were treated with 250 µM palmitic acid at indicated time points; 40 µg of total protein extract was resolved on SDS-PAGE. Immunoblotting was performed using p53 and p21 antibodies, and β-actin was used as a loading control. (**D**) Quantitative analysis of (C) by ImageJ software. (**E**) Palmitic acid, not oleic acid, specifically activated p53 expression. HCT116 cells were treated with 250 µM oleic acid and 250 µM palmitic acid for 24 h. Immunoblotting was performed using p53 and p21 antibodies. (**F**) Quantitative analysis of E by ImageJ software. The data are expressed as mean ± SE of three independent experiments.

**Figure 2 ijms-20-06268-f002:**
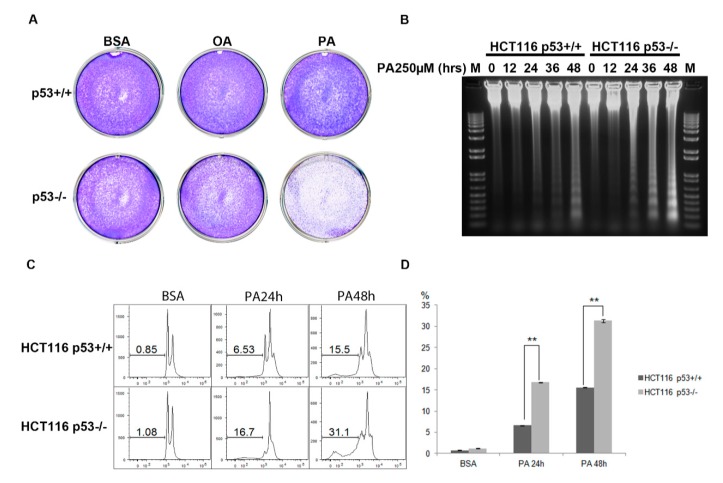
Distinct apoptosis effects induced by palmitic acid in HCT116 p53^+/+^ and p53^−/−^ cell lines. (**A**) HCT116 cells were treated with 250 µM oleic acid or 250 µM palmitic acid for 24 h. Crystal violet staining was performed to show all cells (blue) in the plate. (**B**) After 24 h, HCT116 p53^+/+^ and p53^−/−^ cells were treated with 250 µM palmitic acid for the indicated times. Cells were harvested, and genomic DNA was extracted. A total of 5 µg of DNA was loaded for each sample on a 1% agarose gel to detect DNA laddering. (**C**) Apoptotic cells (Sub-G1) were quantified by flow cytometry analysis after treating with 500 µM palmitic acid for the indicated time. Cells were stained with 2 µg/mL propidium iodide. DNA content analysis was performed. Values indicate the percentage of sub-G1 cells. (**D**) Percentages of apoptotic cells were quantified as the mean ± SE of three independent experiments. Asterisks represent significant differences between HCT116 p53^+/+^ and p53^−/−^ (*p* < 0.01) at indicated PA treatment time points.

**Figure 3 ijms-20-06268-f003:**
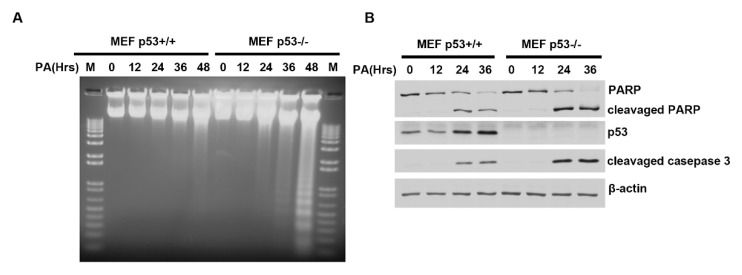
Distinct PA-induced apoptosis effects in mouse embryo fibroblasts (MEF) p53^+/+^ and p53^−/−^ cell lines. (**A**) Primary MEF cells were treated with palmitic acid for the indicated time periods (0–48 h). Genomic DNA was isolated from primary MEF after treatment with palmitic acid. A total of 5 µg of DNA was loaded for each sample on a 1% agarose gel to detect DNA laddering. (**B**) Primary MEF cells were treated with palmitic acid for the indicated time. A total of 40 µg of total protein extract was resolved on SDS-PAGE. p53, cleaved caspase-3, PARP, and cleaved PARP were detected by Western blotting. β-actin was used as a loading control. (**C**) FACS analysis of primary MEF p53^+/+^ and p53^−/−^ cells treated with palmitic acid for the indicated time. Cells were fixed with cold methanol, stained with propidium iodide, and subjected to DNA content analysis by flow cytometry. (**D**) Primary MEF p53^−/−^ cells showed greater apoptosis than primary MEF p53^+/+^ cells following 24 h of palmitic acid treatment, visualized by staining with Annexin V-FITC for 15 min at RT. An Advanced Microscopy Group (AMG) Evos f1 microscope was used for imaging. The scale is 25 µm.

**Figure 4 ijms-20-06268-f004:**
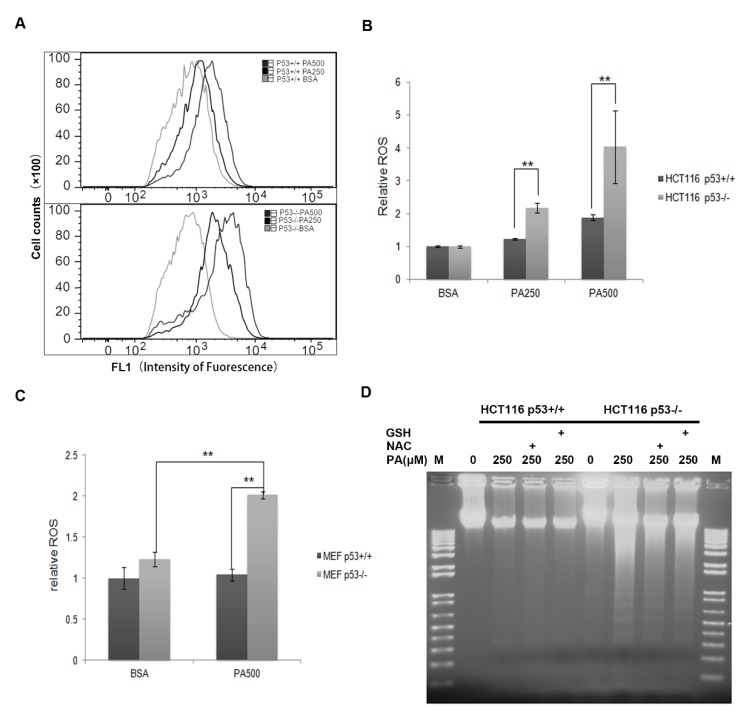
HCT116 p53^−/−^ cells accumulate more reactive oxygen species (ROS) than p53^+/+^ cells after palmitic acid treatment. (**A**) Palmitic acid induces ROS in a dose-dependent manner after 24 h of treatment. ROS were analyzed by flow cytometry as described in Materials and Methods. HCT116 p53^+/+^ and p53^−/−^ cells were treated with 0, 250, and 500 μM palmitic acid for 24 h. (**B**) Quantification of ROS induced by palmitic acid in HCT116 p53^+/+^ and p53^−/−^ cell lines. The data are expressed as mean ± SE of three independent experiments (*n* = 3). Asterisks represent significant differences between HCT116 p53^+/+^ and p53^−/−^ cells (*p* < 0.01). (**C**) Relative ROS levels of primary MEF p53^+/+^ and p53^−/−^ cells. The data are expressed as mean ± SE of three independent experiments (*n* = 3). Asterisks represent significant differences between primary MEF p53^+/+^ and p53^−/−^ cells (*p* < 0.01). (**D**) A 24 h treatment with the antioxidants glutathione (GSH, 20 µg/mL) or N-Acetyl cysteine (NAC, 200 µg/mL) attenuated primary MEF cell apoptosis caused by palmitic acid in MEF p53^−/−^ cells. A total of 5 µg of DNA was loaded for each sample on a 1% agarose gel to detect DNA laddering.

**Figure 5 ijms-20-06268-f005:**
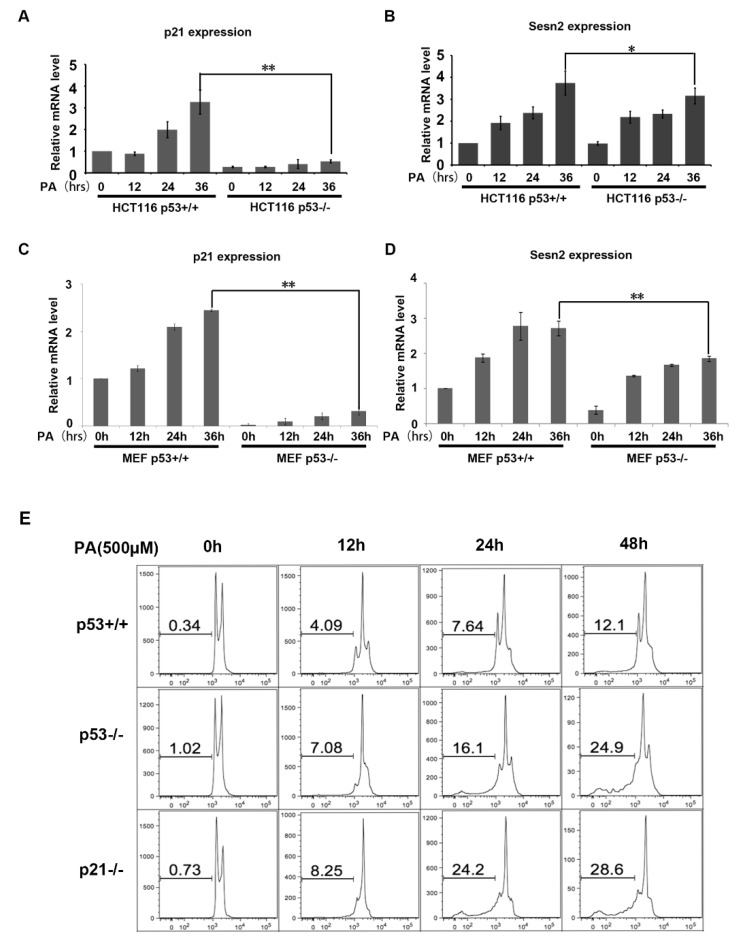
Expression of *p21* and *Sesn2* genes in p53^+/+^ and p53^−/−^ cells under PA stress. (**A**) Relative mRNA expression of *p21* under 250 µM PA stress at indicated time points in HCT116 p53^+/+^ and p53^−/−^ cell lines. ** indicates *p* < 0.01, *n* = 6. (**B**) Relative mRNA expression of *Sesn2* under 250 µM PA stress at indicated time points in HCT116 p53^+/+^ and p53^−/−^ cell lines. * indicates *p* < 0.05, *n* = 6. (**C**) Relative mRNA expression of *p21* under 250 µM PA stress at indicated time points in MEF p53^+/+^ and p53^−/−^ cell lines. ** indicates *p* < 0.01, *n* = 6. (**D**) Relative mRNA expression of *Sesn2* under 250 µM PA stress at indicated time points in MEF p53^+/+^ and p53^−/−^ cell lines. ** indicates *p* < 0.01, *n* = 6. (**E**) Apoptotic cells (Sub-G1) were quantified by flow cytometry analysis after treating with 500 µM palmitic acid for the indicated time in HCT116 p53^+/+^, HCT116 p53^−/−^, and p53 p21^−/−^ cell lines. Cells were stained with 2 µg/mL propidium iodide. DNA content analysis was performed. Values indicate the percentage of sub-G1 cells.

**Figure 6 ijms-20-06268-f006:**
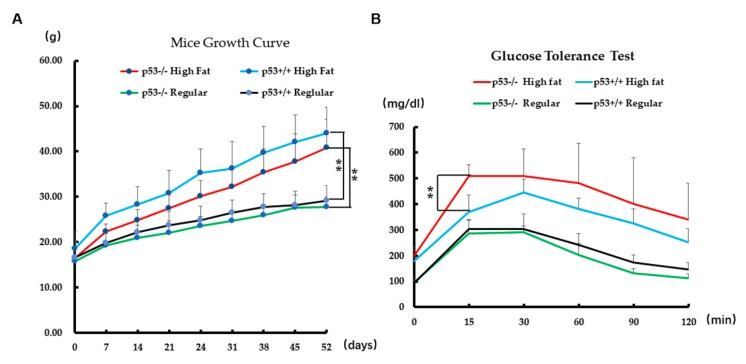
Biological effects of PA stress on mice feed a high fat diet. (**A**) Growth curves of mouse weight under a high fat diet and a regular diet. ** indicates *p* < 0.01, *n* = 5. (**B**) Glucose tolerance test results at indicated time points after glucose injection. ** indicates *p* < 0.01, *n* ≥ 3.

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
