# Peer review of "Loss of p53 Sensitizes Cells to Palmitic Acid-Induced Apoptosis by Reactive Oxygen Species Accumulation"

_ijms, 2019, doi:10.3390/ijms20246268_

Round 1

Reviewer 1 Report

Yu el al. demonstrated that the most common saturated free fatty acid, palmitic acid can induce lipotoxicity, in which the critical role of using p53 was demonstrated by using colon carcinoma cells with or without p53 (HCT116) and MEF isolated from WT and KO mouse. The authors also showed p53 play an important role using obesity mouse model through glucose tolerance test. Therefore, the authors insist that p53 contributes cell apoptosis in response in response to PA treatment and can potentially protect cells from PA-induced apoptosis. Although the authors demonstrated the important role of p53 using various method in vitro and in vivo, there are still not enough experimental evidences to prove the author’s conclusion.

Introduction part generally has the rationale and explanation to help reader’s understating    rather than the description of results.

In the result and discussion, the authors described that role of Sesn2 is important for ROS regulation, it seems that Sens2 expression between cells with p53+/+ and p53-/- are not significantly different in Figure 5B and 5D. Even there is no statistics. Compared to p21 expression (Figure 5A, 5C), supporting data is too weak based on the differences of Sesn2 expression. Based on this, summary of the role of Sesn2 is not acceptable in Figure 7.

Compared to Figure 2D (cell cycle), patter of cell cycle (BSA-treated group) is different from Figure 3C. the cell cycle pattern of MEF p53+/+ is different from other cell types?

The authors tested the concentration and treatment period of PA to induce apoptosis using various methods (Figure 1- Figure 3C). However, 500uM PA was unexpectedly used from Figure 3D. Please explain about this.

First of all, the authors used an obesity model using p53 wt and ko mice to show the role of p53 in lipotoxicity induced by PA treatment. However, glucose tolerance test did not represent the cell death by excessive ROS production by PA (even the protective effect showed only in 15min). As the authors described in discussion, b-cell apoptosis and ROS generation in mouse tissue level should be addressed at least because these data are critical to support the author’s hypothesis (Figure 7).

There are some misinterpretation and grammar errors.

Author Response

Dear Reviewer:

We highly appreciate your valuable comments on our manuscript. The suggestions are quite helpful for us. The manuscript has been revised according to your comments and suggestions, and we look forward to hearing from you regarding our submission.

The responses to your specific comments are as follows:

Comments and Suggestions for Authors

Yu el al. demonstrated that the most common saturated free fatty acid, palmitic acid can induce lipotoxicity, in which the critical role of using p53 was demonstrated by using colon carcinoma cells with or without p53 (HCT116) and MEF isolated from WT and KO mouse. The authors also showed p53 play an important role using obesity mouse model through glucose tolerance test. Therefore, the authors insist that p53 contributes cell apoptosis in response in response to PA treatment and can potentially protect cells from PA-induced apoptosis. Although the authors demonstrated the important role of p53 using various method in vitro and in vivo, there are still not enough experimental evidences to prove the author’s conclusion.

Response: Thank you! We make a revision to cancel the potential model part (Figure 7). In addition, we make a statistic analysis about Figure 5 A, C and D. A new data was added in figure 5 using HCT116 p21-/- cell line treated with PA. We revised the introduction part.

Introduction part generally has the rationale and explanation to help reader’s understating rather than the description of results.

Response: We revised the introduction part to help reader’s understating for my results. 

In the result and discussion, the authors described that role of Sesn2 is important for ROS regulation, it seems that Sens2 expression between cells with p53+/+ and p53-/- are not significantly different in Figure 5B and 5D. Even there is no statistics. Compared to p21 expression (Figure 5A, 5C), supporting data is too weak based on the differences of Sesn2 expression. Based on this, summary of the role of Sesn2 is not acceptable in Figure 7.

Response: Thank you! According to your key comments, we cancel the figure 7. As you mentioned, we did a statistics analysis (n=6)about Sens2 expression between cells with HCT116 p53+/+ and HCT116 p53-/-(in 36 hours). There is the difference in figure 5B treatment time for 36 hours (P=0.0275<0.05). It appears significantly difference between MEF p53+/+ and p53-/- under PA treatment(P<0.01)In addition, in order to further investigate p21’s roles, we add a new data in figure 5E. please see the new figure 5.

Compared to Figure 2D (cell cycle), patter of cell cycle (BSA-treated group) is different from Figure 3C. the cell cycle pattern of MEF p53+/+ is different from other cell types?

Response: Thank you for your careful commences. We focus on sub-G1 apoptotic cells and do not specially notice cell cycle. Only to find that PA will lead to G1/S arrest by p21. We guess that the pattern of cell cycle is different from Figure 3C compared to Figure 2D because different cell types respond differently to BSA.

The authors tested the concentration and treatment period of PA to induce apoptosis using various methods (Figure 1- Figure 3C). However, 500uM PA was unexpectedly used from Figure 3D. Please explain about this.

Response: Thank you! In fact, in order to get high quality picture, we test different dose and concentration of PA, because this picture is best in 500μM PA for 24 hours.

First of all, the authors used an obesity model using p53 wt and ko mice to show the role of p53 in lipotoxicity induced by PA treatment. However, glucose tolerance test did not represent the cell death by excessive ROS production by PA (even the protective effect showed only in 15min). As the authors described in discussion, b-cell apoptosis and ROS generation in mouse tissue level should be addressed at least because these data are critical to support the author’s hypothesis (Figure 7).

Response: Thank you! Yes,glucose tolerance test did not represent the cell death by excessive ROS production by PA. In fact, we try to detect beta-cell apoptosis and ROS generation in mouse tissue, but we failed to do good section of pancreatic tissue. In order to improve our manuscript, we canceled Figure 7.

There are some misinterpretation and grammar errors.

Response: Thank you! We did language processing to revise some misinterpretation and grammar errors

Reviewer 2 Report

This article by Yu et al. investigates the role of p53 in palmitate-induced lipotoxicity using p53 proficient and deficient cell lines. The authors demonstrate higher sensitivity of p53-/- cell lines to palmitate induced cell death, showing increased markers of DNA fragmentation, loss of cell survival, ROS generation, and apoptosis compared to p53+/+ cells. A molecular cascade of events is suggested that involves the p53 targets p21 and Sesn2 to prevent ROS production, caspase activation and cell death by apoptosis. However, further experimental evidence is needed to support this model.

The authors fail to introduce and discuss previous work in the field. The introduction paragraph is too short and contains an inappropriate summary of the results and methods section. The following paper (among others) should be cited and brought into context with the present work:

Palmitate induces VSMC apoptosis via toll like receptor (TLR)4/ROS/p53 pathway.

Atherosclerosis. 2017 Aug;263:74-81. doi: 10.1016/j.atherosclerosis.2017.06.002. Epub 2017 Jun 3.

The role of ROS is unclear in the present work. Does it act up- or downstream of p53? The authors propose that ROS induces the expression of p53, p21 and Sesn2, however, experimental support for this is missing.

- How does NAC or other ROS scavengers affect p53, p21 and Sesn2 expression? This experiment should be performed.

- Data presented in Fig. 4D is not very convincing and lacks quantification and information on reproducibility. Other markers used in this study (DNA fragmentation using flow cytometry, immunoblotting, etc.) should be used to corroborate and quantify the effect of NAC.

The following paper should also be cited and discussed in this context:

ROS-Mediated p53 Induction of Lpin1 Regulates Fatty Acid Oxidation in Response to Nutritional Stress.

Mol Cell. Volume 44, Issue 3, 4 November 2011, Pages 491-501

Further support of the proposed model would be desirable. Does knockdown of p21 and/or Sesn2 also sensitize cells to PA stress (and thus mimick the p53-/- condition)? Alternatively, can overexpression of p21 and/or Sesn2 protect cells from loss of p53?

The in vivo mouse data is potentially interesting. It would be nice to link the rest of the work to the in vivo Is p53, p21 and Sesn2 induced in, for instance, pancreas tissue? Does HFD induce beta-cell apoptosis in p53-/- mice? Food intake of mice should be measured? Can the differences in body weight be explained by altered food intake?

Minors:

1 and others lack statistics. Instead of stating “more serious diabetes” (which is an exaggeration) (page 9, line 3), I suggest to correctly state that “glucose tolerance was reduced” or alike. PI co-staining should be used in Fig. 3D to confirm the apoptotic phenotype 3B lacks quantification and replicates. It seems as if there is no difference in the timing (onset) of PARP cleavage and caspase 3 activation in p53 -/- compared to p53 +/+ cells (in contrast to the results obtained with markers of DNA fragmentation). How does PA concentration affect PARP and caspase 3 cleavage? (e.g. using immunoblots from experiments shown in Fig. 1) The title of the work is somewhat misleading. It is unclear what the authors refer to with “distinct cell apoptosis effect”. I suggest to change the title to “… loss of p53 sensitizes cells to PA-induced apoptosis…” or alike Regarding the animal experiments, an ethics approval number should be provided. Moreover, were males or females used for the experiments? Were mice fasted overnight or during day time? Please provide the exact numbers of mice used per group. (unfortunately, 3 mice is a very low number for in vivo experiments). The authors state on page 13, line 7 that they performed an “intraperitoneal glucose test”, but refer to “gavage” in line 10 of the same paragraph. Clearly more detailed and correct information is needed for the animal experiments!

Author Response

Dear Reviewer:

We highly appreciate your thoughtful comments on our manuscript. The suggestions are quite helpful for us. The manuscript has been revised according to your comments and suggestions, and we look forward to hearing from you regarding our submission. Please see the attachmental file.

Round 2

Reviewer 1 Report

The authors modified and improved the manuscript. However, there are still some minor points to review.

In abstract, the last sentence should be modified without ‘mouse’ as the authors already mentioned in rebuttal letter.

And the authors performed apoptosis analysis using Annexin V staining in Figure 5E. This staining method does not have PI staining either? In that case, could you display another general way?

Author Response

Dear Reviewer:

Thanks for your valuable comments! The manuscript has been revised according to your comments and suggesteions, and we look forward to hearing from you regarding our submission.

In abstract, the last sentence should be modified without ‘mouse’ as the authors already mentioned in rebuttal letter.

Resoponse: Thank you! We have revised the manuscript to cancel the mice.

And the authors performed apoptosis analysis using Annexin V staining in Figure 5E. This staining method does not have PI staining either? In that case, could you display another general way? 

Resoponse: Thank you! We used PI staining to detect Sub-G1 cells. The method is the same as Figure 2D which describe in 4.6 part in materials and methods.
